Prediction of the spread of African swine fever through pig and carcass movements in Thailand using a network analysis and diffusion model

http://orcid.org/0000-0003-2968-9182 Poolkhet Chaithep 1 fvetctp@ku.ac.th
http://orcid.org/0000-0001-7462-6514 Kasemsuwan Suwicha 1
Thongratsakul Sukanya 1
Warrasuth Nattachai 2
Pamaranon Nuttavadee 2
http://orcid.org/0000-0003-2790-1100 Nuanualsuwan Suphachai 3 4
1 Veterinary Public Health, Kasetsart University , Kamphaeng Saen, Nakhon Pathom , Thailand
2 Department of Livestock Development, Ministry of Agriculture and Cooperatives , Bangkok , Thailand
3 Department of Veterinary Public Health, Chulalongkorn University , Bangkok , Thailand
4 Center of Excellence for Food and Water Risk Analysis (FAWRA), Faculty of Veterinary Science, Chulalongkorn University , Bangkok , Thailand
Hussein Mona
Electronic publication date: 2023 May 9
Publication date: 2023
Volume: 11
Electronic Location ID: e15359
Received 2022 Aug 4; Accepted 2023 Apr 16
Copyright: © 2023 Poolkhet et al.
Copyright year: 2023
Copyright holder: Poolkhet et al.
License: This is an open access article distributed under the terms of the Creative Commons Attribution License, which permits unrestricted use, distribution, reproduction and adaptation in any medium and for any purpose provided that it is properly attributed. For attribution, the original author(s), title, publication source (PeerJ) and either DOI or URL of the article must be cited.
License URL: https://creativecommons.org/licenses/by/4.0/

Keywords: African swine fever (ASF), Carcasses and products, Diffusion model, Network analysis, Pig, Thailand

Funding: Agricultural Research Development Agency (Public Organization) of Thailand PRP6105022740 This work was supported by the Agricultural Research Development Agency (Public Organization) of Thailand (Grant No. PRP6105022740). The funders had no role in study design, data collection and analysis, decision to publish, or preparation of the manuscript.

==============================
Background

African swine fever (ASF) is a serious contagious viral disease of pigs that affects the pig industry. This study aimed to evaluate the possible African swine fever (ASF) distribution using network analysis and a diffusion model through live pig, carcass, and pig product movement data.

Material and Methods

Empirical movement data from Thailand for the year 2019 were used, and expert opinions were sought to evaluate network properties and the diffusion model. The networks were presented as live pig movement and carcass movement data at the provincial and district levels. For network analysis, a descriptive network analysis was performed using outdegree, indegree, betweenness, fragmentation, and power law distribution, and cutpoints were used to describe movement patterns. For the diffusion model, we simulated each network using spatially different infected locations, patterns, and initial infection sites. Based on expert opinions, the initial infection site, the probability of ASF occurrence, and the probability of the initial infected adopter were selected for the appropriated network. In this study, we also simulated networks under varying network parameters to predict the infection speed.

Results and Conclusions

The total number of movements recorded was 2,594,364. These were divided into 403,408 (403,408/2,594,364; 15.55%) for live pigs and 2,190,956 (2,190,956/2,594,364; 84.45%) for carcasses. We found that carcass movement at the provincial level showed the highest outdegree (mean = 342.554, standard deviation (SD) = 900.528) and indegree values (mean = 342.554, SD = 665.509). In addition, the outdegree and indegree presented similar mean values and the degree distributions of both district networks followed a power-law function. The network of live pigs at provincial level showed the highest value for betweenness (mean = 0.011, SD = 0.017), and the network of live pigs at provincial level showed the highest value for fragmentation (mean = 0.027, SD = 0.005). Our simulation data indicated that the disease occurred randomly due to live pig and carcass movements along the central and western regions of Thailand, causing the rapid spread of ASF. Without control measures, it could spread to all provinces within 5- and 3-time units and in all districts within 21- and 30-time units for the network of live pigs and carcasses, respectively. This study assists the authorities to plan control and preventive measures and limit economic losses caused by ASF.

Introduction

African swine fever (ASF) is a serious contagious viral disease in pigs that can spread globally. Previous ASF outbreaks resulted in massive losses in pig production that caused great economic losses in the infected countries. For example, China, with the world’s highest pig production, experienced an ASF outbreak that resulted in an estimated economic loss of approximately US$ 141 billion (OIE, 2020). ASF is caused by the African swine fever virus (ASFV), a DNA arbovirus that belongs to the family Asfarviridae and genus Asfivirus. ASFV causes African swine fever (ASF), a fatal disease of domestic as well as wild pigs. Warthogs, bushpigs, and giant forest hogs act as reservoir hosts of ASFV (Dixon, Sun & Roberts, 2019; OIE, 2021). ASF transmission from infected to healthy animals can occur directly or indirectly through the insect vector, Ornithodoros sp., commonly known as soft ticks (OIE, 2021). However, no official reports exist on the presence of soft ticks associated with ASF in Thailand. The incubation period for ASF ranges from 3 to 19 days. Clinical signs of ASF infection can be peracute, acute, or subacute and include chronic conditions such as fever, anorexia, depression, vomiting, diarrhoea, abortion, skin haemorrhage, or even death in severe cases. Currently there is no effective treatment or vaccine available for ASF (Calkins & Scasta, 2020; OIE, 2021). ASF has a 30% to 70% mortality rate, which can reach 100% in the case of domestic pigs. Therefore, a study examining the nature of this disease in areas that may differ in terms of geographic and risk factors, including forecasting the likelihood of disease occurrence and spread, is important to develop effective disease control strategies.

ASF was first reported in Kenya in 1921 (Cisek et al., 2016). Subsequently, the disease continued to spread across Africa, Europe, and Asia (Sargsyan et al., 2018; OIE, 2021). Previous studies attribute the spread of ASF to several risk factors. ASF spread during 2007–2012 in Russia was attributed to anthropogenic, farm biosecurity, and spatial factors (Oganesyan et al., 2013). In the Baltic countries and in Poland, the prevalence of ASF peaked in wild boars due to hunting activities in winter (EFSA et al., 2017). In Romania, proximity to pig farms with ASF outbreaks was an important risk factor attributing ASF spread to commercial and backyard farming (Boklund et al., 2020). In China, pig density on farms is an important predictor of ASF outbreaks (Ma et al., 2020). There have been instances of cross-border ASF spread facilitated by transportation of contaminated pork and pig products to other countries from China (Vergne et al., 2017; Kim et al., 2019). ASF spread has also been attributed to other anthropogenic risk factors such as road network density (Gulenkin et al., 2011) and pigs and pig products trading (Lichoti et al., 2016; Petrini et al., 2019).

Several mathematical models have been formulated to predict the occurrence and spread of ASF. The basic reproduction number (R0) was estimated using infectious disease modelling (Gulenkin et al., 2011; Korennoy et al., 2017) and experimental data (de Carvalho Ferreira et al., 2013; Guinat et al., 2016) of an ASF epidemic or transmission. Barongo et al. (2015) used various mathematical models and reported that R0 ranged from 1.58–3.24. A semi-quantitative model for estimating the risk of ASF spread in Europe through wild boars helped identify Finland, Romania, Latvia, and Poland as high-risk regions and identified the most influential risk factors for the spread of ASF to be wild boar habitat and outbreak density (de la Torre et al., 2015). Qualitative risk assessment using a hierarchical process indicated that endemic areas had a moderate risk in the Trans-Caucasus countries and Russia and a high chance of spreading the disease in other areas (Wieland et al., 2011). Spatial multi-criteria decision analysis (MCDA) was used to predict the area of suitability for ASF. The researchers found that areas that contained wildlife predominantly presented a sylvatic transmission cycle. Furthermore, areas with close contact among pigs predominantly exhibited a domestic transmission cycle (de Glanville et al., 2014). An outbreak simulation was performed for the transmission of ASF through animal movements; ASF occurred in approximately 1.84% of the total pig population in the United States (Podgórski & Śmietanka, 2018). The dynamics of the spread of ASF within domestic pigs were simulated, and indirect transmission between farms within a radius of 2 km was the predominant mode of transmission (Mur et al., 2018). The diffusion model has been previously used for predicting the spread of infectious diseases using mathematical models (Elkin, Topal & Bebek, 2017; Kumar & Sinha, 2021). For network data, the diffusion model can be used to study the complexity of the network structure and dynamic changes by times (Kumar & Sinha, 2021; Moody, 2002).

In Thailand, prior to the start of this study in 2019, no official report on the occurrence of ASF existed to support the claim that Thailand was free from ASF. Predicting disease spread using mathematical models can facilitate effective control measures for ASF. This study aimed to evaluate and predict ASF spread using a network analysis and diffusion model using live pig, carcass, and pig product movement data. In addition, the results of the study can also facilitate the determination of possible patterns in the spread of ASF in other countries.

Materials and Methods

Study framework and ethical statements

Movement data from 2019 were used, and expert opinions were sought to evaluate the movement network properties and the diffusion model (Fig. 1 shows the research process in this study). The level of evaluation was described in epidemiological units at the provincial and district levels. Movement data were obtained from the Department of Livestock Development (DLD) in Thailand. These data were recorded in the DLD database under the name ‘e-movement.’ The data were divided into two categories: (1) movement data of pigs, piglets, and wild pigs, referred to in the current study as ‘live pig movement.’ The total number of provinces and districts was recorded as 77 (in 2019, Thailand was divided into 77 provinces) and 861 (in 2019, Thailand was divided into 878 districts), respectively; (2) movement data of carcasses, offal, pork, and products of pig origin, such as bacon, ham, and sausages, collectively referred to as ‘carcasses and pig products movement.’ The total number of provinces and districts was 77 and 873, respectively. Each movement data of live pigs and carcasses was classified into the following: origin of movement, destination of movement, identification number, date, type of animal/products, and purpose of movement.

Figure 1 Schematic diagram of descriptive network analysis and diffusion network analysis used to predict the spread of African swine fever through pig and carcass movements in Thailand.

A diffusion model was used to predict the possibility of ASF spreading patterns and important parameters were estimated based on expert opinions. Before network parameter suggestions were received from the experts, the results of the movement data were shown at a meeting. After the meeting, experts from the meeting along with a few new experts who did not attend the meeting were randomly selected. The 25 experts selected were specialists from the pig industry from the DLD, university, and private sectors; their network parameter suggestions were obtained using Google Forms. Other questions related to the ASF situation in Thailand, effective control measures, preventive methods, and the probability of ASF occurrence through live pigs and/or carcasses were also included.

This study was conducted in accordance with the principles of the Declaration of Helsinki. Data from the DLD were approved for analysis, interpretation, and publication with official permission. In addition, email approval was received from all experts to use, analyse, and release their opinions and the Google form without publishing their names.

Network analysis

A descriptive network analysis of live pigs and carcass movements was performed using directional and valued networks (Borgatti, Everett & Johnson, 2018). Table 1 lists the network centralities used in this study for all networks. Centrality values are presented as outdegree, indegree, betweenness, and fragmentation to describe movement patterns. Undirected degree centrality was used to calculate the power law distribution. Cutpoints and components were used to describe subgroup properties. All calculations in the descriptive network analysis were performed using Ucinet 6.675 software (Analytic Technologies, Lexington, KY, USA). Provinces and districts with the highest centralities were displayed with ArcGIS release 10.2 (ESRI, Redlands, CA, USA).

Table 1 Description of the parameters used for the descriptive network analysis to predict the spread of African swine fever through pig and carcass movements in Thailand.

Parameter	Description	
Outdegree	This parameter is a centrality estimator in the network. It evaluates individual actors (province/district) to determine the number of provinces or districts for sending pigs or carcasses to other actors. A normalization value has been used in this study.	
Indegree	The normalized centrality value of the network to evaluate individual actors that determine the number of provinces or districts for obtaining pigs or carcasses from other actors.	
Degree	The centrality value of the network to measure the number of connections of individual actors with others, regardless of the direction of pig or carcass transport. In this study, the degree centrality value was used to create a histogram to assess movement patterns of pigs and carcasses.	
Betweenness	Network normalized centrality value used to evaluate the shortest paths of individual actors for every pig or carcass transport event.	
Fragmentation	Individual network centrality value used to estimate the proportion of any pair of actors that cannot reach each other.	
Cutpoint	The property of actors that splits the network into subgroups when this actor is removed.	
Component	In this study, this subgroup level analysis was used to estimate the property of weak and strong components. In addition, a giant component is also presented, which indicates the subgroup consisting of the most connected actors in the network.	
Power law	The exponential correlation of degree values that reflects the phenomenon where a small number of actors is clustered. This cluster probably influences other actors.	

Diffusion model

A diagram of the diffusion model is presented in Appendix 1. We simulated each network using different spatially infected locations and patterns, represented as marginal (border of the network), centre (centre of the network), and random. Based on expert opinion, the initial infection site was a marginal location for live pigs and a random location for carcasses. According to the experts estimation of input parameters, the transmission probability of ASF through live pigs and carcasses was 0.5 and 0.36, respectively. The probability of the initial infected adopter was 0.5 and 0.05 at the provincial and district level, respectively. The probability values of initial infected adopters varied using 0.1, 0.5, and 0.7 for provincial networks and 0.01, 0.05, and 0.1 for district networks. We used 30 time-steps to include the movement data for each network. The simulation output under varying network parameters was captured in a line graph with 50 time-steps to estimate and compare the infection speed. The software used for simulation was R 3.6.2 (R Core Team, Vienna, Austria), and the line graph was plotted using Microsoft Excel Release 2201 (Microsoft, Redmond, WA, USA). In addition, the statistical significance was determined using Moran’s I at an alpha error of 0.05.

Results

General information

From the live pig and carcass movement data in 2019, the total number of movements recorded was 2,594,364. These were divided into 403,408 (403,408/2,594,364; 15.55%) for live pigs and 2,190,956 (2,190,956/2,594,364; 84.45%) for carcasses moved. Therefore, carcass movements were approximately 5.43 times greater than live pig movements.

Descriptive network analysis

The live pig and carcass movement data were subjected to network analysis. The live pig and carcass movements in districts and provinces were represented as ‘nodes’ and the pairs of districts as well as provinces were called ‘ties’. Network centrality values such as outdegree, indegree, betweenness, and fragmentation were determined to describe movement patterns of live pigs and carcasses (Table 2). The mean outdegree and indegree values for each parameter were identical (Table 2). The details of the network centrality results in each province are presented in Appendix 2. Network centrality results with respect to district level data are presented in Appendix 3. The nature of movement patterns (random or scale-free) under each network was determined based on whether or not they followed a power law function.

Table 2 Descriptive statistics of network parameters of live pigs and carcasses movements used to predict the spread of African swine fever in Thailand.

Statistics	Network parameters of live pigs at provincial level (n = 77)	Network parameters of live pigs at district level (n = 861)	Network parameters of carcasses and pig products at provincial level (n = 77)	Network parameters of carcasses and pig products at district level (n = 873)	
Out1	Ind2	Bet3	Frag4	Out	Ind	Bet	Frag	Out	Ind	Bet	Frag	Out	Ind	Bet	Frag	
Mean	56.060	56.060	0.011	0.027	0.529	0.529	0.002	0.003	342.554	342.554	0.010	0.026	2.832	2.832	0.001	0.003	
SD5	82.670	84.708	0.017	0.005	1.406	1.881	0.007	0.001	900.528	665.509	0.018	0.002	21.868	5.579	0.005	0.003	
Min6	0	0.040	0	0.013	0	0	0	0	0	1.040	0	0.013	0	0	0	0	
Max7	513.790	645.066	0.086	0.052	16.228	42.371	0.111	0.012	4,649.803	5,626.092	0.111	0.039	385.649	53.390	0.103	0.041	
Notes:

1 Out = outdegree.

2 Ind = indegree.

3 Bet = betweenness.

4 Frag = fragmentation.

5 SD = standard deviation.

6 min = minimum.

7 Max = maximum.

Network analysis of provincial live pig movement data

In 2019, the number of ties generated for live pig movement in the 77 provinces was 1,794. A prominent tie appeared in the connection between Ratchaburi and the Nakhon Pathom provinces. Table 2 presents the results of network analysis that indicated the outdegree, indegree, betweenness, and fragmentation have a mean 56.060 (standard deviation; SD = 82.670), 56.060 (SD = 84.708), 0.011 (SD = 0.017), and 0.027 (SD = 0.005), respectively. The means were similar for the outdegree and indegree values. The indegree maximum value was higher than the outdegree value. The Ratchaburi and Nakhon Pathom (Fig. 2) provinces showed the highest outdegree and indegree values, respectively. The betweenness and fragmentation values were low with a few variations. This means that there was no predominant province when considering these values. The Nakhon Ratchasima and Nakhon Pathom (Fig. 2) provinces showed the highest values for betweenness and fragmentation, respectively. No province exhibited the cutpoint property. The provinces were connected as weak single components with a giant strong component of size 98.7%. The degree distribution of this network did not follow power laws (Fig. 3A), which indicates that live pig movements in the provinces followed a random pattern.

Figure 2 Map of Thailand shows the highest value of outdegree (Out), indegree (Ind), betweenness (Bet), and fragmentation (Frag) of live pig movements (LPM) and carcass movements (CM).

Source: Map generated by ArcGIS 10.2 software.

Figure 3 A network graph of live pigs at the provincial level (A), and district level (B), network of carcasses at the provincial level (C), and district level (D) were tested for a power-law distribution.

Network analysis of district live pig movement data

At the district level, out of the 861 nodes, we found 11,775 ties in the network. Descriptive network analysis results of the mean (and SD in brackets) of the network centrality values represented by outdegree, indegree, betweenness, and fragmentation were 0.529 (1.406), 0.529 (1.881), 0.002 (0.007), and 0.003 (0.001), respectively (Table 2). The mean outdegree and indegree values were lower than that of the live pig network at the provincial level. The SDs of the outdegree and indegree values were approximately two to four times greater than the mean values (Table 2). This is an indication that the variability in each district was high. A prominent tie was observed between the Chom Bueng district of Ratchaburi and the Muang district of Nakhon Pathom. The maximum outdegree and indegree values (Table 2) indicate that there was a high frequency of pigs moving out or into the districts. The Pak Tho district of Ratchaburi (Fig. 2) had the highest frequency of live pigs moving out. The Muang district of Nakhon Pathom (Fig. 2) had the highest frequency of live pigs moving in. The betweenness and fragmentation values were low with a few variations, as is the case of the provincial live pig movement network. This implies that there is no predominant district when considering these values. The Muang district of Nakhon Pathom and the Pak Chong district of Nakhon Ratchasima (Fig. 2) had the highest values of betweenness and fragmentation. Cutpoint events were identified in 62 nodes, which accounted for 7.06% of the nodes (Appendix 3). A single weak component, which was contained in a strong giant component with a size of 73.4%, was observed. The degree distribution trends followed power laws (Fig. 3B), indicating that the network was likely to be scale-free.

Network analysis of provincial carcass movement data

The provincial carcass movement network comprised 77 nodes with 1,682 ties. Compared with the live pig movement network at the provincial level, it was found that the frequencies of moving live pigs and carcasses were 403,408 and 2,190,956, respectively. In other words, the movement of carcasses was 5.43 times more frequent than that of live pigs. A prominent tie was observed between the Nakhon Pathom and Bangkok provinces. The mean (SD) of the network centrality values of outdegree, indegree, betweenness, and fragmentation were 342.554 (900.528), 342.554 (665.509), 0.010 (0.018), and 0.026 (0.001), respectively (Table 1). The mean outdegree and indegree values were similar. The outdegree and indegree values in this network were the highest when compared to the other networks (Table 2). Chachoengsao province had the highest outdegree value, while Bangkok had the highest indegree value (Fig. 2). The SDs of the outdegree and indegree values were approximately three and two times greater than the mean, respectively. The betweenness and fragmentation values (including SD) were low, as were the live pig movement networks. The Nakhon Pathom and Chiang Mai provinces had the highest values of betweenness and fragmentation (Fig. 2). There were no cutpoints in this network. A single weak component containing a strong component with a size of 98.7% was observed. Many inconsistencies were observed in the power law distribution in this network (Fig. 3C), which indicates that this network exhibited a random pattern.

Network analysis of district carcass movement data

The district carcass movements comprised 873 nodes with 7,830 ties. A prominent tie was observed in the connection between the Muang district of Nakhon Pathom and the Bangkok Noi district of Bangkok. The mean (and SD in brackets) of network centrality, represented by outdegree, indegree, betweenness, and fragmentation were 2.832 (21.868), 2.832 (5.579), 0.001 (0.005), and 0.003 (0.003), respectively. The mean outdegree and indegree values were lower than that of the carcass network at the provincial level. The SDs of the outdegree and indegree values were approximately 10 and two times greater than the mean values, respectively. Most districts (top 20) adjacent to Bangkok had high outdegree values (Appendix 2). The highest outdegree value was that of the Muang district of Chachoengsao (Fig. 2). The district with the highest indegree value in this network was Muang of Samut Sakhon (Fig. 2). The Muang district of Nakhon Pathom and the Phunpin district of Suratthani had the highest betweenness and fragmentation values, respectively (Fig. 2). Cutpoints were observed in 27 (27/878; 3.07%) nodes, which accounted to 3.07% of the nodes in the network (Appendix 3). A large weak component that contained a strong component with a size of 41.0% was oberved. In this network, the degree distribution followed power laws (Fig. 3D) which indicated that this could be a scale-free network.

Diffusion model

This model was based on the hypothesis that Thailand had no control measures for ASF. Infection spread was predicted based on movement data and the initial number of adopters (infected provinces and districts).

Diffusion model of live pig movement

The diffusion model predicted that all provinces could be infected within 5-time units in the case of live pig movement at the provincial level (Table 3). The highest infection adopters were observed at the first time-step, where 38 out of 77 provinces (49.35%) could get infections (Table 3). In the case of live pig movements at the district level, all districts could get infected within 21-time units. The highest infection adopters were observed at time-step 9, where 163 out of 861 districts (18.93 %) could become infected (Table 3).

Table 3 Results of network prediction using a diffusion model to predict the spread of African swine fever through pig and carcass movements in Thailand.

Time step	Live pigs at provincial level (n = 77)	Live pigs at district level (n = 861)	Carcasses at provincial level (n = 77)	Carcasses at district level (n = 873)	
AD1	CAD2	CP3	AD	CAD	CP	AD	CAD	CP	AD	CAD	CP	
1	38	38	0.49	43	43	0.05	38	38	0.49	43	43	0.05	
2	19	57	0.74*	9	52	0.06*	29	67	0.87	43	86	0.10*	
3	15	72	0.94*	7	59	0.07*	10	77	1	39	125	0.14*	
4	4	76	0.99	12	71	0.08*	0	77	1	30	155	0.18*	
5	1	77	1	44	115	0.13*				31	186	0.21*	
6	0	77	1	68	183	0.21*				39	225	0.26*	
7				108	291	0.34*				31	256	0.29*	
8				149	440	0.51*				37	293	0.34*	
9				163	603	0.70*				28	321	0.37*	
10				114	717	0.83*				38	359	0.41*	
11				68	785	0.91*				40	399	0.46*	
12				30	815	0.95*				68	467	0.53*	
13				14	829	0.96*				62	529	0.61*	
14				5	834	0.97*				58	587	0.67*	
15				2	836	0.97*				38	625	0.72*	
16				9	845	0.98*				28	653	0.75*	
17				2	847	0.98*				33	686	0.79*	
18				3	850	0.99*				24	710	0.81*	
19				4	854	0.99*				18	728	0.83*	
20				6	860	1				8	736	0.84*	
21				1	861	1				17	753	0.86*	
22				0	861	1				15	768	0.88*	
23										20	788	0.90*	
24										15	803	0.92*	
25										15	818	0.94*	
26										10	828	0.95*	
27										4	832	0.95*	
28										6	838	0.96*	
29										6	844	0.97*	
30										1	845	0.97*	
Notes:

1 AD = adopters (infected province or district).

2 CAD = cumulative adopters.

3 CP = cumulative probability.

* P < 0.05 (please note that the software does not report the exact P-value).

Diffusion model of carcass movement

In the case of carcass movement, the diffusion model predicted that all provinces could be infected within 3-time units, which is faster than that of live pig movements (Table 3). The highest infection was predicted to occur in the first time-step where 38 out of 77 provinces (49.35%) could get infected (Table 3). ASF-spread predictions at the district level showed that all districts could get infected in approximately 30-time units (Table 3). The highest infection was predicted to occur at the 12th time-step, where 68 out of 873 districts (7.79%) could get infected (Table 3).

Diffusion model with parameter changes

Based on the 2019 live pig and carcass movement data of provinces and districts, the input parameters of the diffusion model were modified to predict the transmission patterns. The simulated results of ASF occurrence are shown in Fig. 4.

Figure 4 Results from a simulated network using different input parameters.

Marginal (m), centre (c), random (r) represented spatially infected locations. The probability of the initial infected adopter was simulated as 0.1 (1), 0.5 (5), and 0.7 (7) for the provincial live pigs (A) and carcasses network (C). For the district network, the probability of the initial infected adopter was changed and compared by 0.01 (01), 0.05 (05), and 0.1 (1) in a live pig (B) and carcasses network (D).

Diffusion model with parameter changes for provincial live pig movement: the highest ASF transmission speed was observed when the probability of the initial infected adopter was kept as 0.7 (Fig. 4A; black line 9). The spatial location here was random. Similar ASF transmission speed was observed in the central (Fig. 4A; brown line 8) and marginal locations (Fig. 4A; blue line 7; the line is completely under the line of central location). When the probability of the initial infected adopter was changed to 0.5, the ASF transmission speed was highest for random locations (Fig. 4A; green line 6), followed by the central and marginal locations (Fig. 4A; blue line 5 and orange line 4, respectively). The slowest ASF transmission speed was observed when the probability of the initial infected adopter was kept at 0.1, but the spatial location behaviour was similar to that of the 0.7 and 0.5 probabilities with the highest speed observed at random locations (Fig. 4A; line 3), followed by central and marginal locations (Fig. 4A; lines 2 and 1, respectively).

Diffusion model with parameter changes for district level live pig movement: ASF transmission speed was the fastest when the probability of the initial infected adopter was set to 0.05 with a random location (Fig. 4B; green line 6). However, when the probability of the initial infected adopter was set to 0.1 with a random location, the ASF transmission speed was faster than the 0.05 random location line in the early stages (Fig. 4B; black line 9).

Diffusion model with parameter changes for provincial carcass movement: The simulation results of carcass movements at the provincial level (Fig. 4C) were almost the same as those of the diffusion model of provincial live pig movements. Highest ASF transmission speed was observed with a probability of the initial infected adopter of 0.7 with a random location (Fig. 4C; black line 9).

Diffusion model with parameter changes for district level carcass movement: The simulation of district level carcass movements (Fig. 4D) showed the highest ASF transmission speed with the probability of the initial infection adopter of 0.1 with a random location (Fig. 4D; black line 9).

Discussion

A descriptive social network analysis (SNA) was used to analyse the provincial and district level movement networks of live pigs and carcasses. Outdegrees and indegrees had the highest values in the network of carcasses at the provincial level. Betweenness and fragmentation values of live pig and carcass provincial networks were higher than those of district networks. In each network, the outdegree and indegree mean values were identical. The degree distribution of both district networks followed a power law function.

The diffusion network analyses were carried out with the assumption that Thailand had no control measures for ASF in place. The simulation results showed that all provinces could be infected within five- and three-time units for the network of live pigs and carcasses, respectively; all districts could be infected within approximately 21- and 30-time units for the network of live pigs and carcasses, respectively. Simulation using different input parameters predicted that a rapid spread of ASF would be facilitated by random patterns of disease occurrence. These results suggest that ASF infection has the potential to spread rapidly if disease control measures are not implemented in the early stage of infection, particularly when disease occurrence follows a random pattern. In this study, the unit of time was calculated according to the steps involved in moving pigs from the source node to the destination node. In some cases, movement may take hours, such as in the case of moving pigs or carcasses between neighbouring districts. In other cases, it may take a day, such as in the case of moving pigs or carcasses between provinces. The results of this study have implications for disease control, studying the principle of maximum likelihood is necessary with respect to worst-case scenarios. Previous studies report that the rapid spread of ASF poses a considerable risk (Sánchez-Vizcaíno, Mur & Martínez-López, 2012; Schulz et al., 2019) which is consistent with the findings of this study.

In 2019, the DLD observed that the number of carcass movements was 5.43 times higher than the number of live pig movements. This indicates that the likelihood of the spread of ASF is higher in carcass movements than in live pig movements. This makes slaughterhouses one of the most important locations for disease surveillance. Previous studies have observed that slaughterhouses are epidemiological endpoints of the spread of ASF (Andraud et al., 2019; Kim, Park & Kang, 2021). In central Thailand, including Bangkok, the presence of many large slaughterhouses contributes significantly to the high indegree values of many provinces and districts for live pig networks, and the high outdegree values of many provinces and districts for the carcass network. Thus, it seems slaughterhouses might play an important role in disease spread in terms of ASF occurrence through live pig movements and disease transmission through carcass movement. However, most slaughterhouses have implemented good biosecurity protocols, including passive surveillance activities such as recording systems for notifiable diseases. Therefore, slaughterhouses are less likely to permit ASF infected pigs or allow the release of ASF infected pig products. Nevertheless, the relevant agencies should implement additional surveillance measures in slaughterhouses, especially during ASF outbreaks in neighbouring countries. Another issue to be addressed would be the movement data logged into the e-Movement database, which cannot be traced back from meat or pig products to live animals. If products can be traced back to live animals and their movements, it would strengthen the existing surveillance protocol. This would also help with quarantine and/or culling measures once the location of live pigs was identified.

At the district level, the network of live pigs and carcass movements followed the power law function, indicating scale-free networks, whereas the networks at the provincial level followed a random pattern. This indicates that the carcass network has a hub at the district level (Barabási & Bonabeau, 2003). To identify the hub, relevant authorities should focus on nodes with high outdegree and/or indegree values. In the case of ASF occurrence in Thailand, the nodes with high outdegree and indegree values must implement surveillance activities and/or implement effective control measures immediately. This would help reduce the size of the epidemic (Prell, 2012; Borgatti, Everett & Johnson, 2018).

Although each network’s outdegree and indegree values were identical, the high SD values of these parameters indicate a high variation of movement data in certain provinces and districts. For example, the Ratchaburi province, which showed the highest outdegree for live pig movement, represents a high-risk province where rapid spread of ASF is possible. These results were correlated with the live pig network at the district level, which indicated that, of the top five districts, the two districts of Ratchaburi had a high outdegree value. The Nakhon Pathom province (with the highest indegree of live pig movements) is likely to encounter a few infected pigs entering the region. This is correlated with the live pig network at the district level, which indicated that Muang of Nakhon Pathom had the highest indegree value. In the case of an outbreak in Thailand, the nodes discussed above need to implement control measures to limit disease spread. Based on the analysis, Ratchaburi and Nakhon Pathom were the most influential areas of entry and exit of infected pigs. We propose that an area with a predominant outdegree needs to implement active surveillance measures, while an area with a predominant indegree needs to implement passive surveillance measures.

All four networks showed low betweenness and fragmentation values with minimal variation, and this indicates that all nodes in all these networks were similar. In the case of ASF outbreaks, the nodes with the highest betweenness and fragmentation values had minor effects on network breakdown compared to the remaining nodes. However, infectious disease control requires a combination of measures, and we suggest setting up animal checkpoints in the nodes with high betweenness/fragmentation values (e.g., at the main road in Nakhon Ratchasima and Nakhon Pathom) to inspect pigs or carcasses. Betweenness is a powerful centrality for measuring nodes and ties and indicates a node’s position within the network. This differs from the indegree/outdegree values, which give more importance to each node and its ties (Prell, 2012). With respect to fragmentation, a previous study reported that the probability of tuberculosis infection was reduced in more fragmented networks. This is because a fragmented node tends to decrease the number of contacts with other nodes (Sintayehu et al., 2017). The implementation of control measures in nodes using betweenness and fragmentation would help reduce the epidemic size and make a difference, even if marginal.

Cutpoints in the networks at the provincial level were not detected. This is because provincial networks lack biconnected components (Fegley & Torvik, 2013). At the district level, many cutpoints were found in networks of both live pig movements (7.06%) and carcass movements (3.07%). Nodes with cutpoints keep the network connected (Wasserman & Faust, 1994). Therefore, if control measures are implemented on these nodes, the infectious network can be disconnected, which means that the magnitude of the disease can be reduced. For component analysis of all networks, the size of the giant strong component within the weak component was relatively large. The proportion of strong giant components in a weak component ranged from 41% to 98.7%. This is the most complicated area in each network, which is correlated with epidemic size (Rautureau, Dufour & Durand, 2011). The presence of a giant strong component probably leads to a small-world network and power-law distribution (Christley et al., 2005b). These reasons are consistent with the results of the diffusion model where ASF spread in Thailand is very rapid. Therefore, a good surveillance system is important for prevention and control of ASF. We propose that the government must invest in the budget to build an effective surveillance system.

The diffusion model with the four parameter inputs showed that the random pattern of ASF occurrence could lead to a rapid spread of infection. Compared to a small-world network, a previous study concluded that the probability of infection was greater in a random network (Christley et al., 2005a). Moreover, the spread of infection can increase dramatically with a few randomly sampled nodes (Gopalan et al., 2011). This suggests that early detection is essential for disease prevention and control, and this can be achieved by implementing surveillance in nodes with high outage values.

Different epidemiological units might present different results. We suggest that network analysis must be performed using data from the smallest epidemiological unit. The smallest epidemiological unit used in this study was at the district level. Movement data was available even at the sub-district level, but analysis with such large data that includes all epidemiological units would require a robust, high-performance computerized system with sophisticated algorithms. This study is limited in that it only takes into consideration the movement data at the district and province levels; the sample size was small although sufficient to generate simulation data for ASF predictions (Fig. 4).

This study was based on data from live pigs and carcass movements that appeared in the e-Movement database of the DLD. Other pig-farming systems, such as domestic and backyard farming, were not included in this study. Therefore, in the event of an outbreak, disease control measures need to cover the otherwise unidentified sectors. It is necessary for relevant authorities to develop a robust database that records all livestock sectors for further analysis and/or modelling with more accuracy and precision. In addition, some of our data is based on expert opinions which may be subject to uncertainty. However, during our study, ASF occurrence was not reported in Thailand. Therefore, expert consultation was necessary for predicting the status of ASF in Thailand.

Conclusions

This study reports the results of a network analysis and simulated diffusion model for ASF in Thailand. We believe that our simulation model will help the authorities identify and prioritize key nodes to contain the spread of ASF. This would subsequently limit economic losses caused by ASF. Our simulation model can also be applied to movement data in other countries to predict and prevent the spread of ASF.

Supplemental Information

Supplemental Information 1 Diagram of diffusion model.

Click here for additional data file.

Supplemental Information 2 Centrality values.

Click here for additional data file.

Supplemental Information 3 List of cutpoints.

Click here for additional data file.

Supplemental Information 4 R code.

Click here for additional data file.

Supplemental Information 5 Questionnaire.

Click here for additional data file.

Supplemental Information 6 Questionnaire in Thai.

Click here for additional data file.

Supplemental Information 7 Raw data from questionnaire.

Click here for additional data file.

Supplemental Information 8 Raw data.

Click here for additional data file.

We thank the experts for their valuable support. We would also like to thank the Agricultural Research Development Agency (Public Organization) of Thailand and the Faculty of Veterinary Science at Chulalongkorn University for providing computerized systems.

Abbreviations

ASF African swine fever

ASFV African swine fever virus

DLD Department of Livestock Development

MCDA multi-criteria decision analysis

R0 basic reproduction number

SD standard deviation

SNA social network analysis

Additional Information and Declarations

Competing Interests

Author Contributions

Data Availability

The authors declare that they have no competing interests.

Chaithep Poolkhet conceived and designed the experiments, performed the experiments, analyzed the data, prepared figures and/or tables, authored or reviewed drafts of the article, and approved the final draft.

Suwicha Kasemsuwan conceived and designed the experiments, authored or reviewed drafts of the article, and approved the final draft.

Sukanya Thongratsakul performed the experiments, prepared figures and/or tables, authored or reviewed drafts of the article, and approved the final draft.

Nattachai Warrasuth conceived and designed the experiments, authored or reviewed drafts of the article, and approved the final draft.

Nuttavadee Pamaranon conceived and designed the experiments, authored or reviewed drafts of the article, and approved the final draft.

Suphachai Nuanualsuwan conceived and designed the experiments, authored or reviewed drafts of the article, project Leader, and approved the final draft.

The following information was supplied regarding data availability:

The code and raw data are available in the Supplemental Files.

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
