# Peer review of "Prediction of the spread of African swine fever through pig and carcass movements in Thailand using a network analysis and diffusion model"

_PeerJ, doi:10.7717/peerj.15359_

## Round 0.1 · original submission · Minor Revisions

Dear author
Please find attached the reviewers' suggestions to be addressed in the next version of your manuscript.

Reviewer 1 ·

Basic reporting

i) The first paragraph of the introduction (lines 47-56) does not set the paper's tone. For example, it does not mention what the authors did in this paper and why? Instead, the first paragraph should act as a hook so that the reader gets interested in reading further.

ii) Line 100: Is the movement data publicly available? Can the source/web link be referenced? It could help future readers.

iii) It is not clear which particular diffusion model it is. A model diagram with inputs, outputs, and parameters would be helpful. In addition, a citation to the paper/work that introduced the diffusion model would be beneficial.

iv) Line 137: The meaning of 'marginal' is unclear to the readers. A small description/definition would be helpful.

v) Line 139: Typically, it can either be a transmission probability or a transmission rate. Please clarify if this is not a mistake.

vi) The information in figure 1 can be tabulated or written in text. I recommend using a high-level diagram of the model used in this paper (see point iii).

vii) Figure 3: Proper axis labeling should be used.

viii) Figure 4: It is unclear what kind of data the plots are showing. The authors should use axis labels and describe those labels as necessary.

ix) Table 1: Outdegree description: 'sources' is a confusing term in the second line of the description. Did the authors intend to write 'destination'?

x) Table 1: Degree description: Clarity can be improved. For example, the degree is the number of connections of individual actors with others, regardless of the direction of pig/carcass transport.

xi) Table 1: Betweenness description: The definition is unclear to a reader.

xii) It is perplexing why the Max / Min values of In and Out degrees are fractional and not integers. Could the authors clarify which formulas were used to compute these values?

Experimental design

no comment

Validity of the findings

no comment

Additional comments

no comment

Reviewer 2 ·

Basic reporting

The manuscript by Poolkhet et al. consists of a comprehensive work integrating African swine fever distribution data and movement data of live pigs, carcasses and pork products.

Overall, it is an interesting and promising work.

Below are some considerations.

Experimental design

At the very beginning of the introduction, the authors mention that this virus is transmitted by a vector, ticks of the genus Ornithodoros. However, there is no further mention of this fact in the text or in the analyses. I wonder if this vector is equally distributed throughout the region studied. There is data on the distribution of the vector, as well as using the distribution of pigs?

Continuing with the hosts in general, are there other relevant hosts that can be found in the area and serve as reservoirs?

Validity of the findings

In any case, since pigs can also be infected directly, my previous comment does not diminish the relevance of the work presented. I do think that they should discuss it at least once, and mention whether data on the distribution of vectors is available.

Additional comments

The resolution of some of the figures seems suboptimal. I suggest to improve the quality and to increase the font size (e.g., fig 2 and 3).

·

Basic reporting

Minor edits required. See report

Experimental design

Minor edits required. See report

Validity of the findings

See report

Additional comments

Reviewer report
Manuscript Number: #75982
Manuscript title: Prediction of the spread of African swine fever through pig and carcass movements in Thailand using a network analysis and diffusion model

Summary:
The study uses network analysis and a diffusion models to evaluate the African swine fever distribution in Thailand. It addresses a key disease in the global swine industry and hence important to the readership of peer J and manuscript is generally well written. However, the write can be improved upon as described below in my comments. Majorly, the reliance on expert opinion to parametrize that model casts some doubt on the reliability of the model findings. Sensitivity and other analyses ought to be done to clear this doubt.

Major revisions/comments:

1. Rewrite the entire abstract so that the reader can easily pick out the separate sections of the abstract i.e. background, materials and methods, results and conclusion. Also explicitly indicate the mean values and other relevant numbers in the abstract.
2. Some few more relevant citations need to be considered in the introduction and discussion sections.
3. You overly base your parameter estimation on expert opinion as you state in line 112. This is a major limitation to this study. You need to test the effect of using such expert elicitation on your study findings and acknowledge that among your study limitations. Such elicitations are never free of bias and hence findings are rarely reproducible. Devote a section to explaining the impact of this and also justifying why you chose this approach when literature exists on some of the parameters such as ASF transmission probabilities.
4. Justify the choices of 30 and 50 time steps in M&M lines 143-144.

Minor revisions:
5. Line 31: Explicitly name the country from which this that is obtained for clarity of the abstract.
6. Lines 31-32: You write “Our networks were presented as the network of live pig movements at the provincial and district level and the network of carcass movements at the provincial and district level.” and there is a lot of repetition. Rewrite this sentence compactly.
7. Line 51. Add citation(s) for the statement “The incubation period for ASF ranges from 3 to 19 days”. Also note that the number given may be slightly different from those by WOAH (see https://www.woah.org/fileadmin/Home/eng/Animal_Health_in_the_World/docs/pdf/Disease_cards/AFRICAN_SWINE_FEVER.pdf).
8. Lines 51-54: Add citation(s) for the clinical signs.
9. Lines 71-74: The citations listed do not necessary match the content mentioned about Ukraine. Consider rewriting the entire sentence and add the respective reference to where they are appropriate. Also on R0 and ASF modelling using field data, Field data based studies such as Barongo et al. Plos One 2015: 10(5):e0125842 cannot be easily overlooked due to the variety of methods used. Consider reading and perhaps citing such literature.
10. Line 95: You write “Movement data from 2019 was used..” but “data” is plural and I suggest you change was to were.
11. Line 106: insert “to” after referred.

---

## Round 0.2 · accepted · Accept

From a scientific point of view, your manuscript represents a significant contribution to the prediction of Africa’s Swine Fever (ASF) spread in Thailand by creating a simulation model using live pig, carcass, and pig movement data. The study has a high degree of novelty and uses modern methods of network analysis and diffusion models to predict ASF spread in Thailand. Based on the reviewed manuscript and on my revision to the re-submitted manuscript, I noticed that the authors have addressed all of the reviewers' comments. I appreciate your revision and re-submission of a well-revised version of the manuscript. Congratulations on accepting your study to be published in PeerJ journal.

Reviewer 1 ·

Basic reporting

The authors have improved the manuscript and provided adequate explanations for the issues I reported earlier. I have nothing further to add.

Experimental design

no comment

Validity of the findings

no comment

Additional comments

no comment